# AI Tools in Glioma Studies on Cell Cultures

## Abstract

Cancer research presents a highly technological field of medicine demanding integration of experimental molecular biology data and computational knowledge. Glioma is one of the most aggressive and heterogeneous forms of brain cancer, characterized by high mortality rates and complex tumor biology. Artificial intelligence (AI) and machine learning (ML) technologies have emerged as transformative tools in glioma research, particularly in the analyzing of protein interaction networks, cell culture models, and clinical outcomes. Traditional diagnostic and research approaches rely on manual morphological analysis and time-consuming genetic screening, limiting the speed of patient stratification and therapeutic development. Current computational approaches enable rapid classification, mutation detection, phenotypic analysis, and survival prediction, accelerating both basic and clinical research in neuro-oncology. We discuss trends in computational modeling of brain cancer using AI tools.

## 1 Introduction

Artificial intelligence (AI) refers to a host of computational algorithms that is becoming a major tool capable of integrating large omics databases in biomedicine. AI is reshaping pharmacology by shortening discovery timelines, potentially reducing attrition, and expanding the design space of therapeutic candidates (Dharmasivam et al., 2026).

Glioblastoma multiforme (GBM), also known as grade IV astrocytoma, is one of the most common and aggressive subtypes of primary brain tumors affecting the glial cells of the brain (Rončević et al., 2025). GBM is the most aggressive and common primary brain malignancy in adults, characterized by poor prognosis and treatment resistance. Despite advancements in treatment options, the median survival is roughly 15 months, underlining the need for novel and effective treatments (Tambi et al., 2025). It is characterized by a high degree of heterogeneity, meaning that although these tumors may appear morphologically similar, they often exhibit distinct clinical outcomes. By associating specific molecular fingerprints with different clinical behaviors, high-throughput omics technologies (e.g., genomics, transcriptomics, and epigenomics) have significantly advanced our understanding of GBM, particularly of its extensive heterogeneity proposing a molecular classification for the implementation of precision medicine (Morello et al., 2025). New experimental data on glioma cell lines provide basis for ML applications (Tsherniak et al., 2017; Langenberg et al., 2025).

AI-based algorithms have demonstrated promising potential in enhancing diagnostics through imaging analysis, radiomics, and tumor segmentation. These technologies could enable non-invasive molecular profiling and early detection of GBM (Rončević et al., 2025).

Thorough exploration of AI tools utilization in GBM-omics to uncover different aspects of GBM (subtype classification, prognosis, and survival) would have a significant impact on both researchers and clinicians, allow drug repositioning (Tasci et al., 2025). We used bioinformatics tools for the computational reconstruction glioma gene networks (Iarema et al., 2023;Turkina et al., 2023).

AI has the potential to redefine the landscape in neuro-oncology and can enhance glioma detection, imaging segmentation, and non-invasive molecular characterization better than conventional diagnostic modalities through deep learning-driven radiomics and radiogenomics (Evangelou et al., 2025).

Here we review main trends in AI tools applications for glioma research and cancer studies, discuss perspectives and challenges related to ML and explainable AI.

Table 1: Lis of top 10 genes associated with glioma

| GENE | DESCRIPTION |
|------|-------------|
| TP53 | tumor suppressor, frequently mutated in gliomas |
| PTEN | phosphatase that suppresses PI3K/AKT signaling |
| EGFR | epidermal growth factor receptor, amplified in glioblastoma |
| PDGFRA, PDGFRB | platelet-derived growth factor receptors |
| AKT1 | serine/threonine kinase in PI3K/AKT/mTOR pathway |
| MTOR | mechanistic target of rapamycin, key growth regulator |
| CDK4, CDK6 c | yclin-dependent kinases driving cell cycle |
| RB1 | retinoblastoma protein, tumor suppressor |

## 2 GENE NETWORK RECONSTRUCTION FOR GLIOMA

Computational reconstruction of gene networks (gene-protein network, or regulatory network, as well as proptein-protein interaction network) is an important step in modeling (Barcelos et al., 2025). Protein-protein interaction (PPI) data are taken from IntAct (Panneerselvam et al., 2023), MINT (Molecular INTeraction) (Zanzoni et al., 2002), BioGRID (Oughtred et al., 2019), DIP (Database of Interacting Proteins) (Xenarios et al., 2000), HPRD (Human Protein Reference Database) (Prasad et al., 2009) databases. Core Tools for gene network reconstruction are Cytoscape (Shannon et al., 2003), STRING (Snel et al., 2000; Szklarczyk et al., 2023), with more specific bioinformatics tools PINA, MTGO, NetBox (Du et al., 2021; Du et al., 2025).

Cytoscape (Shannon et al., 2003) is open-source platform for PPI visualization, clustering, and topological analysis of the networks (e.g., degree centrality for hub genes like EGFR in GBM) (https://cytoscape.org/). Plugins like CytoHubba identify key nodes; ClueGO handles enrichment (Lotia et al., 2013; Chin et al., 2014). The STRING tool generates PPI networks with confidence scores, integrates GBM expression data from TCGA (The Cancer Gene Atlas). Cytoscape StringApp imports and analyzes large networks serving as standard for data presentation (Szklarczyk et al., 2023). Series of studied considered Gene Regulatory Networks reconstruction particularly in gliomas (Ladha et al., 2010; Clarke et al., 2015; Affolter et al., 2025). We have reconstructed gene network of glioma (Babenko et al., 2017; Gubanova et al., 2021; Dergilev et al., 2022) to analyze hub genes as possible gene targets. Computer reconstruction of gene networks and current challenges were discussed in (Dergilev et al., 2021; Klimontov et al., 2023).

Computer analysis of associative gene networks using literature mining is a promising approach for interactome analysis (Demenkov et al., 2011; Ivanisenko et al., 2020; Ivanisenko et al., 2025).

Analysis of gene expression profiles and network allows find significant prognostic biomarkers and potential therapeutic targets in GBM (Beygi et al., 2026). Table 1 presents the 10 genes associated with glioma. Note, that this list was reconstructed using AI tool String-Chat (https://string-db.org) based on LLM model.

Figure 1 shows the interactions of known human genes related to glioma progression – TP53, PTEN, KRAS (reconstructed by STRING-DB).

Recently, new tools for gene network reconstruction were developed. Coletti and colleagues (Coletti et al., 2025) designed a two-step framework for variable selection using sparse network estimation across various omics datasets. This framework incorporates MINGLE (Multi-omics Integrated Network for GraphicaL Exploration), a novel methodology designed to merge distinct multi-omics information into a single network, enabling the identification of underlying relations through an innovative integrated visualization.

Note Gene Association/ML-Driven Networks – PINA, MTGO, NetBox:

PINA (Protein Interaction Network Analyzer): Builds non-redundant PPI networks, filters for GBM-specific modules, and infers tumor-type expression specificity with TCGA integration (Du et al., 2021; Du et al., 2025; Cowley et al., 2012).

MTGO: Identifies topological/functional modules in GBM PPI networks, comparing disease vs. healthy for disrupted pathways.

NetBox: Automates module detection in GBM mutation networks, highlights core pathways like PI3K/p53.

Recently, Barcelos and co-authors analyzed transcriptional data from 989 primary gliomas in The Cancer Genome Atlas (TCGA) and the Chinese Glioma Genome Atlas (CGGA) (Barcelos et al., 2025). GRNs were reconstructed using the RTN package which identifies regulons - the sets of genes regulated by a common transcription factor (TF) based on co-expression and mutual information. Regulon activity was evaluated through Gene Set Enrichment (GSEA). Elastic net regularization and Cox regression identified 31 and 32 prognostic genes in the TCGA and CGGA datasets, respectively, with 11 genes overlapping, many of which are associated with neural development and synaptic processes. GAS2L3, HOXD13, and OTP demonstrated the strongest correlations with survival outcomes (Barcelos et al., 2025).

## 3 MACHINE LEARNING TOOLS IN GLIOMA STUDIES

Machine learning models have revolutionized the analysis of glioma cell cultures through multiple complementary approaches. Deep learning convolutional neural networks (CNNs) extract morphological parameters from microscopic images of glioblastoma cell cultures, enabling researchers to identify cell behavior patterns and predict responses to therapeutic interventions without manual annotation. These image-based models can quantify proliferation rates, migration dynamics, and cellular morphology with unprecedented precision and consistency. Hollon and colleagues (2023) developed DeepGlioma, a rapid (¡90 seconds), artificial-intelligence-based diagnostic screening system to streamline the molecular diagnosis of diffuse gliomas (Hollon et al., 2023). Genomic abnormalities associated with primary GBMs include mutations in EGFR, PTEN, NF1, TERT, PIK3R1, and CDKN2A-p16INK4a genes whereas those that are common to secondary GBM are in IDH1/2, TP53, and PDGFR genes. In addition to these, several studies have indicated how germline (inherited) and somatic (acquired) mutations poses potential vulnerability of glioma development in an individual over time. Reports indicate that several genes are frequently associated with germline mutations in GBM, with NF1, TP53, and APC being the most mutated genes in GBM (Tambi et al., 2024).

Using diverse types of omics data, many AI tools primarily machine learning tools have been developed in the past decades to understand different aspects of glioblastoma (Mohammadzadeh et al., 2025).

Genetic analysis of glioma cultures has been accelerated through AI-driven mutation detection (Park et al., 2024). Machine learning platform DeepSomatic detects cancer-causing DNA variants across sequencing platforms with dramatically improved accuracy, having been successfully applied to pediatric glioblastoma samples. Explainable machine learning models can classify major glioma subtypes—including astrocytoma, oligodendroglioma, and glioblastoma—from RNA-sequencing data while generating interpretable predictions of patient survival outcomes. Computer vision pipelines coupled with 3D microtumor assays represent another innovative application. These systems analyze individual patient tumor responses to various anti-cancer treatments in real time, providing personalized drug sensitivity profiles that guide therapeutic selection. Such approaches bridge in vitro cell culture research and clinical practice by maintaining three-dimensional tumor architecture while leveraging AI for high-throughput phenotypic analysis.

Karakas et al. (2025) predicted the IDH1 genotype in gliomas using radiomics and machine learning (ML) methods. indicate that radiomic analyses provide comprehensive genotypic classification by assessing the entire tumor and present a safer, faster, and more patient-friendly alternative to traditional biopsies. The study highlighted the potential of radiomics and ML techniques, particularly KNN, Ensemble, and SVM, as powerful tools for predicting the molecular characteristics of gliomas and developing personalized treatment strategies (Karakas et al., 2025).

A systematic review and meta-analysis were conducted to assess the performance of AI-based models in predicting survival outcomes for HGG patients (Mohammadzadeh et al., 2025). A total of 39 studies with 29 various algorithms and 79,638 patients were included, with 15 studies contributing to the meta-analysis. The most commonly used algorithms were random forest (RF) and logistic

regression (LR), which demonstrated robust predictive accuracy. The pooled AUCs for one-year, two-year, three-year and overall survival predictions were 0.816, 0.854, 0.871 and 0.789 respectively. Subgroup analysis revealed that RSF achieved the highest predictive accuracy with an AUC of 0.91, while LR followed with an AUC of 0.89. Models integrating clinical, radiomics, and genetic features consistently outperformed single-data-type models.

Six machine learning models-XGBoost, Random Forests, Support Vector Machines, Artificial Neural Networks, Extra Trees Regressor, and K-Nearest Neighbors - were employed to classify patients into predefined survival categories (Onciul et al., 2025). XGBoost demonstrated the highest predictive accuracy, achieving a mean ROC-AUC of 0.90 and an accuracy of 0.78. Ensemble models outperformed simpler classifiers, emphasizing the predictive value of metadata.

## 4 KEY AI MODELS IN GLIOMAS

AI models are increasingly used to predict drug responses in glioma cell cultures by analyzing transcriptomic data, morphological features, and high-throughput screening results. These tools leverage machine learning and deep learning to forecast sensitivity to therapies like antineoplastics, helping to customize the treatment for glioblastoma (GBM).

Key AI Models:

- NeurixAI: This neural interaction explainable AI model predicts tumor-specific drug responses from transcriptomic profiles and drug properties, using data from 476 cancer cell lines (including glioma) treated with 1,135 drugs. (Keyl et al., 2025). It standardizes the log AUC for growth inhibition and excels in modeling GBM viability post-treatment. NeurixAI takes model data from repositories like DepMap (Corsello et al., 2020). PRISM, and GDSC, imputing responses for new patient-derived glioma cell lines even across tissue types. For instance, NeurixAI maps tumor gene expression (about 19,000 genes) to drug vectors for precise log AUC predictions.

- Hybrid CNN-LSTM-Transformer Model (Naveed et al., 2026) is a deep learning architecture that integrates convolutional neural networks (CNN), long short-term memory (LSTM), and transformers to predict drug resistance in glioblastoma cell lines, outperforming prior models on GDSC datasets. HybridDeepSynergy leverages CNNs to capture local feature interactions, LSTMs to model sequential dependencies, and attention mechanisms to extract long-range relationships within the data.

- Predictive ML on PRISM Screens (Corsello et al., 2020). Machine learning algorithms trained on high-throughput screening of 64 GBM patient-derived cell cultures forecast sensitivity to 280 antineoplastic drugs, clustering samples by histological type and validating predictions experimentally.

GBM Feature Selection ML Pipeline: Uses SHAP (SHapley Additive exPlanations) explanations on GDSC (Genomics of Drug Sensitivity in Cancer) data to select features distinguishing GBM cell lines, predicting drug sensitivities, and identifying repurposing candidates like pathway-targeted agents. Shi and colleagues (2025) developed interpretable machine learning (ML) model based on multicenter magnetic resonance imaging (MRI) radiomics data for predicting the World Health Organization (WHO) grade in glioma patients (Shi et al., 2025).

Bioimage Analysis with YOLO

YOLO (You Only Look Once) models, known for real-time object detection, have been adapted for bioimaging to identify different states in biological objects (Alhwaiti et al., 2025; Chen J. et al., 2025).

Chourib I. (2025) proposed a robust and scalable deep learning framework for brain tumor detection and classification, built upon an enhanced YOLO-v11 architecture combined with a two-stage transfer learning strategy. The first stage involves training a base model on a large, diverse MRI dataset. Upon achieving a mean Average Precision (mAP) exceeding 90 percents, this model is designated as the Brain Tumor Detection Model (BTDM). In the second stage, the BTDM is fine-tuned on a structurally similar but smaller dataset to form Brain Tumor Detection and Segmentation, effectively leveraging domain transfer to maintain performance despite limited data. The model is further optimized through domain-specific data augmentation-including geometric transformations-to improve generalization and robustness (Chourib, 2025).

## 5   Key Challenges of AI in Cell Cultures Studies

AI tools have fundamentally transformed glioma cell culture research, enabling rapid morphological analysis, genetic characterization, and treatment response prediction. As these technologies continue to mature and integrate with emerging multimodal analysis approaches, they will accelerate the transition from basic research discoveries to personalized clinical therapeutics for glioma patients (Evangelou et al., 2025). Future development should focus on improving model interpretability, expanding dataset diversity, and standardizing analytical pipelines across research institutions.

Artificial intelligence is reshaping pharmacology by shortening discovery timelines, potentially reducing attrition, and expanding the design space of therapeutic candidates. Alongside technical milestones, regulatory and ethical frameworks from the US Food and Drug Administration and European Medicines Agency are beginning to address transparency, bias, accountability, intellectual property, and data privacy (Dharmasivam et al., 2026).

To reconstruct gene networks, cognitive systems for automatic text mining of scientific publications and databases are often employed. One such AI-driven platform, ANDSystem (Associative Network Design), is designed for automatic knowledge extraction of molecular interactions and, on this basis, the reconstruction of associative gene networks (Ivanisenko et al., 2025).

Text analysis and Large language models (LLMs) like ChatGPT 4.0 hold promise for enhancing clinical decision-making in precision oncology. The advent of high-throughput transcriptomic technologies, particularly bulk and single-cell RNA sequencing, coupled with artificial intelligence (AI) methodologies has enabled the development of models capable of capturing high-dimensional gene expression patterns to more comprehensively predict therapeutic resistance (Schmutz et al., 2025; Zhang et al., 2025).

## 6   Key Challenges of AI in Cell Cultures Studies

Thus, the applications of AI tools and methods in glioma research are diverse, having great potential to grow. The application of machine learning to GBM metadata offers a robust approach to predicting patient survival. Machine learning models can integrate multimodal data to develop personalized treatments. They can also enhance prognostication by predicting survival, recurrence, and treatment responses, helping clinicians to make more informed decisions. AI is also revolutionizing pharmacotherapy by identifying novel molecular targets and optimizing combination therapies. Despite notable progress, challenges persist. Limited data quality and quantity, algorithm interpretability, integration problems, and ethical considerations, remain significant challenges to clinical implementation.

### Acknowledgments

The work was supported by Russian Science Foundation.

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
