# OpenReview forum: "AI Tools in Glioma Studies on Cell Cultures"
_mathai.club/MathAI/2026/Conference — MathAI 2026 Conference Submission_

### Official Review · Reviewer_CvvB · 2026-03-11
**The manuscript is presented as a review but lacks methodological rigor and contains substantial bibliographic errors, including unverifiable citations. Therefore, it cannot be recommended for publication in its current form.**

**Rating:** 3
**Confidence:** 4

**Review:**

1. There is no description of the review methodology. It is not specified by what criteria the articles were selected or which protocol was used.
2. The article lists numerous AI tools but does not provide their comparative analysis according to criteria such as: accuracy, computational efficiency, and applicability to different types of data.
3. The review only mentions articles on topics close to the one stated in the title, but does not discuss the features and limitations of the mentioned methods.
4. Issues of reproducibility in the methods mentioned in the review are not addressed: it is unclear whether code, data, or model parameters are available for reproducing the results.
5. The formulations are promotional rather than scientific, for example, "unprecedented precision and consistency". Each statement should be supported by specific examples or references to validated studies.
6. The text states: "Figure 1 shows the interactions of known human genes related to glioma progression...", but the figure is missing.
7. Sections 5–6 duplicate the heading "KEY CHALLENGES OF AI IN CELL CULTURES STUDIES" and are similar in content.
8. There is no mention of the ethical aspects, norms and limitations of the use of AI, which is critically important for the area under consideration.
9. Comments on references:
Alhwaiti Y, Khan M, Asim M, Siddiqi MH, Ishaq M, Alruwaili M. Leveraging YOLO deep learning models to enhance plant disease identification. Sci Rep. 2025 Mar 7;15(1):7969. doi: 10.1038/s41598-025-92143-0. Sci Rep 2025; 15, 7969. doi: 10.1038/s41598-025-92143-0.
Formatting issues: the doi is written twice in one reference.
In addition, the problem of enhance plant disease identification is not related to the stated topic of the review.
10. References cannot be found by doi.
Beygi HS, Shahraki A, Sheervalilou R. Identification of hub genes and signaling pathways as possible therapeutic targets in human glioblastoma: evidenced by bioinformatics analysis. Brain Res. 2026 Mar 1;1874:150137. doi: 10.1016/j.brainres.2025.150137.
The date indicated is 2026 Mar 1, the doi contains 2025, and the article with this doi cannot be found.
11. These articles cannot be found by doi:
Alkan S, et al. Drug Repurposing in Glioblastoma Using a Machine Learning Approach with SHAP Explanations. bioRxiv (PMC12732686). 2025. doi:10.1101/2025.12.07.621456.
Naveed S, Husnain M, Alsubaie N. HybridDeepSynergy: A hybrid deep learning model integrating CNN, LSTM, and attention mechanisms for cancer drug synergy prediction. Comput Biol Med. 2026 Feb 15;203:111471. doi: 10.1016/j.compbiomed.2026.111471.
12. Incorrect citation of sources in the text:
"Despite advancements in treatment options, the median survival is roughly 15 months, underlining the need for novel and effective treatments (Tambi et al., 2025)."
"Reports indicate that several genes are frequently associated with germline mutations in GBM, with NF1, TP53, and APC being the most mutated genes in GBM (Tambi et al., 2024)."
That is, mentions for 2024 and 2025, but in references there is only one entry.
Tambi R, Zehra B, Vijayakumar A, Satsangi D, Uddin M, Berdiev BK. Artificial intelligence and omics in malignant gliomas. Physiol Genomics. 2024 Dec 1;56(12):876-895. doi: 10.1152/physiolgenomics.00011.2024.
13. In the text: "Protein-protein interaction (PPI) data are taken from IntAct (Panneerselvam et al., 2023)", but in the list the year is 2024:
Panneerselvam K, Porras P, Del-Toro N, Perfetto L, Shrivastava A, Ragueneau E, Reyes JJM, Orchard S, Hermjakob H; IMEx Consortium Curators. IntAct Database for Accessing IMEx's Contextual Metadata of Molecular Interactions. Curr Protoc. 2024 Oct;4(10):e70018. doi: 10.1002/cpz1.70018.
14. In the text: "We have reconstructed gene network of glioma (Babenko et al., 2017; Gubanova et al., 2021; Dergilev et al., 2022) to analyze hub genes as possible gene targets. Computer reconstruction of gene networks and current challenges were discussed in (Dergilev et al., 2021; Klimontov et al., 2023)."
There are "Dergilev et al., 2022" and "Dergilev et al., 2021", but in the reference list both entries are from 2021.
15. Two lines merged (18 and 19):
Evangelou K, Kotsantis I, Kalyvas A, Kyriazoglou A, Economopoulou P, Velonakis G, Gavra M, Psyrri A, Boviatsis EJ, Stavrinou LC. Artificial Intelligence in the Diagnosis and Treatment of Brain Gliomas. Biomedicines. 2025 Sep 17;13(9):2285. doi: 10.3390/biomedicines13092285 19. Gubanova N.V., Orlova N.G., Dergilev A.I., Oparina N.Y., Orlov Y.L. Glioblastoma gene network reconstruction and ontology analysis by online bioinformatics tools. Journal of Integrative Bioinformatics.2 021a.Vol. 18, pp. 20210031. doi: 10.1515/jib-2021-0031.
16. There is no reference to the description of the String-Chat tool (LLM model).
17. The article is in the references, but not mentioned in the text:
Islam MT, Yang F, Komladzei S, Akhter M, Sardiu ME, Li Y. Integrative machine learning reveals hidden and emerging co-regulatory gene networks for multi-phase glioblastoma outcome prediction. Eur J Cancer. 2026 Feb 5;234:116197. doi: 10.1016/j.ejca.2025.116197. Epub 2025.